# Extraction, Characterization, and Chitosan Microencapsulation of Bioactive Compounds from *Cannabis sativa* L., *Cannabis indica* L., and *Mitragyna speiosa* K.

**DOI:** 10.3390/antiox11112103

**Published:** 2022-10-25

**Authors:** Srisan Phupaboon, Maharach Matra, Ronnachai Prommachart, Pajaree Totakul, Chanadol Supapong, Metha Wanapat

**Affiliations:** 1Tropical Feed Resources Research and Development Center (TROFREC), Department of Animal Science, Faculty of Agriculture, Khon Kaen University, Khon Kaen 40002, Thailand; 2Department of Animal Science, Faculty of Agriculture and Natural Resources, Rajamangala University of Technology, Tawan-Ok, Chonburi 20110, Thailand; 3Division of Animal Science, Faculty of Agricultural Technology, Rajamangala University of Technology Thanyaburi, Pathum Thani 12130, Thailand; 4Department of Animal Science, Faculty of Agriculture, Rajamangala University of Technology Srivijaya, Nakhon Si Thammarat 80240, Thailand

**Keywords:** *Cannabis*, microencapsulation, bioactive compounds, chitosan encapsulation

## Abstract

The objective of the research was to investigate the bioactive compounds of herbal plant leaves by microencapsulation technique for future application as a feed additive. In this experiment, three herbal plant leaves, namely *Cannabis sativa* L., *Cannabis indica* L., and *Mitragyna speiosa* K., were comparatively investigated using different methods to extract their bioactive compounds. Two methods were used to extract the bioactive compounds: microwave extraction (water-heating transferred) and maceration extraction (methanol extracted). The results obtained using microwave extraction revealed that the total polyphenolic and flavonoid contents and antioxidant capacity were significantly higher and stronger, respectively, than those produced by the maceration extraction method (*p* < 0.05). Furthermore, the spray-drying technique was employed to enhance the extracted compounds by encapsulation with chitosan through ionic gelation properties. The physical characteristics of chitosan-encapsulated substrates were examined under a scanning electron microscope (SEM) and were as microparticle size (1.45 to 11.0 µm). The encapsulation efficiency of the bioactive compounds was found to be 99.7, 82.3, and 54.6% for microencapsulated *M. speiosa*, *C. indica*, and *C. sativa*, respectively. Therefore, microwave treatment prior to chitosan encapsulation of leaf extracts resulted in increased recovery of bioactive compound encroachment.

## 1. Introduction

Many researchers investigated nano/microencapsulation technology to improve stability, enhance bioaccessibility, impart controlled release features, and improve storage and handling convenience. The encapsulation process can potentially enhance production efficiency in the animal feed industry. Harsh environmental factors, including oxygen, heat, pH, and light, can significantly alter the physicochemical characteristics of feed products [1]. Through the entrapment of solid, liquid, or gaseous components as active core materials of polymers or as wall or carrier materials, a multicomponent structure is created. The sizes of the items produced by this method range from 1 to 1000 µm. The numerous morphologies produced by the microencapsulation process, especially the spray-drying technique, include microspheres, microcapsules, and microparticles [2]. The advantages of the spray-drying technique include rapid processing, versatility, cost-effectiveness, ease of scaling up, high encapsulation efficiency (EE), and relatively storage stability [3]. Chitosan [poly (β-(1 → 4)-2-amino-2-deoxy-D-glucose], a cationic polysaccharide or copolymer of chitin, glucosamine, and N-acetyl glucosamine units, can be employed as a wall material. The ionic gelation method has a significant impact on the EEs of the microcapsules, in addition to low cost and not requiring advanced equipment; this method has been used to produce wall material for microencapsulation via spray drying. Furthermore, chitosan has been reported to improved rumen fermentation and mitigate methane production [1,2,4]. Additionally, chitosan microencapsulation has been widely used in applications related to specific food or feed additives to improve biocompatibility, renewable sources, non-toxic and non-allergenic properties, and the ability to biodegrade [5,6].

Numerous factors influence the extraction efficiency of phytonutrients from different herbal plant materials. The separation of bioactive molecules, particularly polyphenolic compounds, such as flavonoids, stilbenoids, and lignans from species of *Cannabis* plants, can be achieved using conventional maceration extraction and advanced microwave extraction methods [7,8]. Microwave extraction can achieve high extraction rates accurate control for the extraction of bioactive compounds. The microwave procedure directly interacts with microwave electrical power to heat the solvent, which causes plant tissue to break down and release active compounds into the normalized solution (water), in addition to a significant increase in internal pressure [7,9,10,11,12]. In addition, maceration extraction has conventionally been used with solvent reagents in several studies, e.g., ethanol, methanol, acetone, ethyl acetate, dichloromethane, and hexane, to explore the extraction of polyphenolic compounds from plants [13,14]. Although conventional extraction methods have advantages, such as simple procedures and operation, they are subject to several drawbacks, including long processing times high heat during extraction, resulting in reduced extraction efficiency and environmental problems affecting humans and farm animals [15].

Due to their widespread use in ruminant animal production, the use of phytochemical components, particularly those from *Cannabis sativa, Cannabis indica*, and *Mytragyna speiosa*, has become a crucial research topic with respect to ruminant nutrition [16,17,18]. In phytochemical studies, researchers have identified bioactive secondary metabolites in phytochemical categories such as condensed tannin, saponin, total phenolic, total flavanol, ∆^9^-tetrahydrocannabinol acid (THCA), and cannabidiol acid (CBDA) components accompanied by high antioxidant capacity in super plants [16,17]. Despite their anti-methanogenic and/or anti-protozoal effects contributing to an increase in ruminal fermentation efficiency, the phytochemicals found in these plants may also improve fiber digestibility and decrease nitrogen excretion. Due to their capacity to be coated with fiber and protein content, phytonutrients affect gas production and reduce CH_4_ production by modifying microbial activities, such as the protozoal population [19,20,21]. Moreover, the continuous release of bioactive compounds in the rumen can considerably improve utilization, resulting in improved animal health.

Therefore, the aim of this study was to investigate extracted bioactive compounds from three plant leaves (*C. sativa*, *C. indica*, and *M. speiosa*) using maceration and microwave extraction. Chitosan was used as a wall material and encapsulated to preserve compounds by the ionic gelation method through a spray-drying process. We also characterized the chitosan microcapsules of bioactive compounds based on their yield of phytonutrient components; determined their morphological characteristics, such as size, shape, and surface structure, under a scanning electron microscope (SEM); and evaluated their encapsulation effectiveness (EE).

## 2. Materials and Methods

### 2.1. Chemicals and Reagents

The following chemicals were procured from Sigma–Aldrich (Sigma-Aldrich™, Burlington, MA, USA): aluminum chloride, sodium acetate, sodium carbonate, potassium persulphate, Folin–Ciocalteu reagent, 2,2-diphenyl-1-picrylhydrazyl (DPPH), 2,2-Azino-bis (3-ethylbenzothiazo-line-6-sulphonic acid) diammonium salt (ABTS), 2,4,6-tripyridyl-s-triazine (TPTZ), ferric chloride hexahydrate (FeCl_3_), methanol, Tween 80, as well as three standards of quercetin (3,3′,4′,5,6-pentahydroxyflavone), gallic acid (3,4,5-trihydroxybenzoic acid), and Trolox (6-hydroxy-2,5,7,8-tetramethylchromane-2-carboxylic acid). Chitosan (Mw = 3.5 × 10^6^) was obtained through a local supplier (solubility in acetic acid: 91.2%, purity ≥ 90%). All chemicals were of analytical reagent grade.

### 2.2. Plant Materials 

The plant samples used in this experiment included *Cannabis sativa* L., *Cannabis indica* L., and *Mitragyna speiosa* M. The plant materials were allocated via cultivation at Khon Kaen University (KKU), Khon Kaen; Rajamangala University of Technology Isan Sakon Nakhon Campus (RUTISKC), Sakon Nakhon; and Rajamangala University of Technology Srivijaya (MUTSV), Nakhon Si Thammarat, Thailand, in greenhouses under controlled management. Approximately 1 kg of fresh leaves (130–140 days of growth) of each plant was carefully harvested and dried at 50 °C for 48 h. Approximately 500 g of the final dry matter (dried leaves) content was ground into a fine powder and stored in a vacuum bag for preservation; samples were later chemically analyzed for essential compounds.

### 2.3. Extraction of Bioactive Compounds 

The dried plant leaf powder was extracted for the bioactive compounds following two methodological procedures: microwave and maceration extraction methods. The step-wise details are illustrated in Figure 1 for methods (1) and (2). The microwave extraction method (MIE) (1) was performed using 30 g of powdered sample extracted with 300 mL deionized water in a microwave power at 100 W for 35 min (final temperature ≤ 60 °C). The maceration extraction method (MAE) (2) was performed as in the preceding experiment but was shaken with methanol solution at room temperature overnight. All supernatant was filtered through cellulose filter paper. Then, the supernatant was collected in a vacuum glass bottle to be analyzed for bioactive compounds and stored at 4 °C for later use in encapsulated formulation processing.

### 2.4. Estimation of Total Polyphenolic and Total Flavonoid Contents and Antioxidant Capacity

Total polyphenolic content (TPC), total flavonoid content (TFC), and antioxidant capacity of the bioactive compound extracts were determined using a 96-well microplate assay via label-free applications (Biochemical assays mode) of an EnSight multimode plate reader (PerkinElmer Inc., Waltham, MA, USA) using Kaleido Data Acquisition software following the protocol described by Phupaboon et al. [22]. 

TPC was measured with Folin–Ciocalteu reagent by absorbance at 765 nm [23]. The TFC was determined based on colorimetric changes with 10% aluminum chloride solution read at 415 nm [24]. All analyses were performed in triplicate, and the results were expressed as mg of gallic acid equivalents (mg GAE) per gram of dry matter (DM) and mg of quercetin equivalents (mg QUE/g DM).

Antioxidant activity was determined using three different methods: the DPPH radical scavenging method [25], ABTS radical scavenging activity [26], and the ferric reducing antioxidant power (FRAP) method [27]. All analyses were performed in triplicate, and the results were expressed as % radical scavenging inhibition and mmol Trolox equivalents (mmol TROE/g DM).

### 2.5. Microencapsulation of Bioactive Compound Formulation Using Spray-Drying Technique

Bioactive extract juice was encapsulated by ionic gelation combined with surfactant ingredients according to the procedure described Nouri [6] and Kurek and Pratap-Singh [2], with slight modifications, as shown in the above procedure of current study. The chitosan 2% (*w*/*v*) wall-material components were dissolved in 1% (*v*/*v*) acetic acid combined with surfactant (2% (*v*/*v*) Tween 80) and gradually stirred at 65 °C until homogeneous. Encapsulation was achieved with wall material loaded with bioactive extract juice in a ratio of 1:1 (*v*/*v*) under constant stirring at room temperature overnight. The microencapsulation of the bioactive compound was spray-dried in a Bǚchi B-191 mini spray dryer. The processes material was fed into a drying chamber using a peristaltic pump operating at a speed of 10 mL/min and a drying airflow of 110 L/h with a pressure drop of 0.73 bar. The inlet temperature was 160 °C, and the outlet temperature was kept at 90 °C. Dried powders were collected, hermetically sealed, and stored at −20 °C until use in in vitro experiments.

### 2.6. Encapsulation Efficiency 

For quantification of encapsulation efficiency (EE), the total polyphenolic content was used as an indirect computational representation of the active ingredient from microcapsules. This method allows for easy reference to a known standard, in this case, gallic acid. The amount of TPC obtained from the initial extraction was compared with the amount of TPC after encapsulation in microparticles released in the inner part of the solvent. The EE was determined by dissolving 1 mg of microparticles in 1 mL of 0.1 N HCl for 48 h, following the method described by Ko et al. [4]. The percentage of entrapment efficiency was calculated based on the amount of bioactive compound (TPC) by an EnSight multimode plate reader (PerkinElmer Inc., USA) at 765 nm [22]. The experiments were performed in triplicate and expressed in % EE according to the following equation [6,28]:(1)$$\mathrm{EE}\;(\%)=\left[\frac{\mathrm{Amount}\;\mathrm{of}\;\mathrm{TPC}\;\mathrm{in}\;\mathrm{extract}}{\mathrm{Amount}\;\mathrm{of}\;\mathrm{TPC}\;\mathrm{in}\;\mathrm{entrapped}}\right]\;\times \;100$$

### 2.7. Morphological Characterization of Microencapsules 

The surface morphology of microparticles was investigated in terms of size, shape, and surface structure of the microcapsules after the chitosan encapsulation process using a field-emission scanning electron microscope (FE-SEM; model: Mira, Tescan Co., Brno, Czech Republic) according to the method described by Ko et al. [4]. 

### 2.8. Statistical Analysis

The results of all data analyses were expressed as mean ± standard deviation (SD) of the triplicate measurements. The obtained results of bioactive compound contents obtained with different extraction methods were statistically analyzed, and the encapsulation efficiencies of three plant species were compared a one-way ANOVA (Duncan’s new multiple range test) via IBM SPSS-KKU Statistics version 27, with *p* < 0.05 indicating a significant difference.

## 3. Results and Discussion

### 3.1. Characteristics of Extracted-Bioactive Compounds

Bioactive compounds were extracted from plant leaves of *C. sativa*, *C. indica*, and *M. speiosa* using two extraction methods: microwave irradiation (with water transferring heat) and maceration (organic solvent extraction), as shown in Table 1. The microwave extraction method achieved a yield of an bioactive compounds from each plant higher than that achieved using the maceration extraction method (*p* < 0.05). This finding is in agreement with the results reported by Yiin et al. [12], who suggested that microwave extraction (MIE) can be carefully controlled based on the temperature and contact time, as well as volumetric heating, as opposed to heat transfer from inside surface inside, which results in a uniform and effective and process. Thus, we analyzed and reported the phytochemical components of TPC and TFC, the antioxidative activities of DPPH, the ABTS radical-scavenging inhibition results, and the FRAP reducing power capacity. The results showed that *M. speiosa* had the highest amount of TPC and TFC at 306.9 mg GAE/g DM and 119.2 mg QUE/g DM when compared with those of *C. indica* and the extract of *C. sativa* extracted using the microwave method (218.9 and 171.7 mg GAE/g DM for TPC and 88.6 and 66.2 mg QUE/g DM for TFC, respectively). Subsequently, the antioxidant activity of three extracted plants was measured as percentage of ABTS inhibition, DPPH inhibition, and the amount of FRAP capacity from the highest to lowest in the ranges of 95.3–43.1%, 91.2–35.0%, and 39.0–14.0 mg/g DM, respectively. Drinić et al. [7] and Matešić [9], reported on the optimization of MAE operating conditions during polyphenolic contents and antioxidant capacity extraction from hemp using varying ratios of ethanol to water and found that the TPC after extraction resulted in the highest DPPH inhibition and FRAP reducing power compared with the maceration method. Several research works investigated the extraction of bioactive compounds from *Cannabis* species via sonication method or ultrasound-assisted extraction (UAE) using different solvents [29,30,31]. In addition, enzyme-assisted extraction techniques (EAEs) with pectinase or cellulose have also been used to extract essential oil or to release phytochemicals from *Cannabis* by increasing the recovery of cannabinoids by 20.2% [32,33]. Although the EASE is highly effective in extracting *Cannabis* phytochemicals, but it is an expensive technique due to the mandatory use of commercial enzymes. Consequently, distilled water and electromagnetic microwave radiation coincide at the time of extraction without impairing the approximate quality of the bioactive ingredients. It is also a simple process for short-term extraction with low cost, making it suitable for further use in the medicinal, therapeutic, and/or feed additive applications. 

In addition, the maceration extraction method was used in this work to extract the bioactive compounds from dried plant leaves with a methanol organic solvent of. The extracted phytochemical content and antioxidant capacity were lower than those obtained with the microwave extraction method (*p* < 0.05). The TPC and TFC were significantly increased following extraction of bioactive compounds from *M. speiosa*, *C. indica*, and *C. sativa*. Furthermore, the antioxidant capacities in terms of DPPH inhibition, ABTS inhibition, and FRAP capacity were also increased, as previously mentioned (Table 1). The main downsides of the organic solvent approach are related to unsuitable materials, such as fats, waxes, and pigments, which dissolve along with the cannabinoids, as well as the extraction temperatures, which frequently cause the destruction of heat-sensitive compounds during processing [8,10,16,34]. In previous studies, selection criteria were based on solvent extraction, such as ethanol or methanol combined with microwave extraction. The results obtained from those extracts were detected in the following ranges: TPC, 0.8499 to 2.7060 mg GAE/mg; TFC, 0.4707 to 1.4246 mg QUE and/or catechin equivalent/mg; antioxidant activity, 0.0009 to 0.2079 mL/mg [11,35,36,37]. Another report indicated that *Cannabis* contained 125 individual cannabinoids, but ∆^9^-THCA and CBDA were most predominant when using ethanol as an effective solvent to extracting these components via hot maceration and through Soxhlet extraction. The extraction efficiency of these two conventional methods was significantly lower than that of the advanced MIE method [8,38]. Therefore, the conventional solvent maceration method achieves inferior results compared to the MIE method. For this reason, in the present study, no other solvents and at varying concentrations were used to extract the bioactive compounds, with a focus on MIE extraction. Furthermore, the effects of microwave extraction processing differ from those of the organic solvent extraction method due to alterations in the cell structure by electromagnetic radiation during extraction [7]. Moreover, advances in microwave extraction have led to the development of cutting-edge processes, including vacuum microwave hydrodistillation, microwave Soxhlet extraction, microwave-assisted Clevenger distillation, etc., which are environmentally friendly methods that can be used to reduce energy use, time required for extraction, solvent use, and waste [10,39,40].

Therefore, we conclude that microwave extraction methods are superior to maceration extraction methods for the extraction of polyphenols and/or antioxidant activity based a comparison of our results with respect to antioxidant activity with those reported in previous research. Herman et al. [41] reported that dried powder of *M. speciosa* leaves was extracted using natural deep eutectic solvent (NADES) via a citric acid-glucose-based MAE method at 270 W. The TPC values of the microwave power were 192.20 mg GAE/g sample, with an extraction time of 15 min and 358.59 mg GAE/g sample. In addition, the antioxidant properties of *M. speciosa* leaf extracts obtained using the MAE method were estimated with DPPH IC_50_ values of aqueous, alkaloid, and methanolic extracts of 213.4, 104.81, and 37.08 µg/mL, respectively [42]. Rezvankhanh [43] reported on optimal condition antioxidant activity with *Cannabis* species of *C. sativa* and/or *C. indi**c**a* obtained from leaves/seed oil extract extracted using the MAE method at 450 W for 7.19 min, with an IC_50_ value of 30.82 mg/mL when compared with the Soxhlet extraction (SE) method, which achieved higher value IC_50_, at 32.47 mg/mL. Isahq et al. [44] investigated *C. indica* extract from leaves, stems, and seeds obtained via maceration extraction using n-hexane to screening the phytochemical components and reported the qualitative presence of alkaloids, saponins, tannins, flavonoids, sterols, and terpenoids.

### 3.2. Bioactive Values and Encapsulation Efficiency of Microencapsulation 

Table 2 shows the formulation of chitosan combined with Tween 80 formulated as a wall material at a ratio of 1:1 ratios and evaluated as the ability to affect the yield (TPC) and (%) EE obtained from bioactive compound extracts. There were significant differences in (%) EE based on the amount of TPC after encapsulation into the wall materials (Table 2). After the encapsulation process TPC values of the bioactive compound extracts were 307.8, 266.1, and 240.0 mg GAE/g DM for encapsulated *M. speiosa*, *C. indica*, and *C. sativa*, respectively. The result of (%) EE for encapsulated *M. speiosa* showed the highest value of 99.7%, followed by 82.3% for encapsulated *C. indica* and 54.6% for encapsulated *C. sativa*. In the current study, a comparison of TPC with (%) EE showed that the correlative significance depends on the polarity difference between the encapsulated wall ingredients and the charged and/or non-charged plant extracts, which are referred to the investigation of high TPC. De Moura et al. [45] reported encapsulation using the ionic gelation property of hydrophilic or low-molecular-weight compounds in the phytochemical structure resulted in problems associated with easy diffusion and rapid release, regardless of the pH, via the ionic gel matrix, which is a drawback and affecting the ability to store important substances. The highest EE (99.7%) of encapsulated *M. speiosa* indicates that it is the best formulation for recovery bioactive compounds in terms of TFC (105.3 mg QUE/g DM), DPPH, and ABTS inhibition (94.8 and 90.3%), with an FRAP capacity of 34.4 mg TROE/mg DM. In addition, encapsulated *C. indica* and *C. sativa* were found to have a radical-scavenging inhibition in the range of 53.0–84.3% more than the efficiency of TFC (69.8 to 22.6 mg QUE/g DM) and FRAP capacity (30.9 to 17.2 mg TROE/g DM) (Table 2). The results of the bioactive values in the initiation and post encapsulation stages did not differ due to the increased retention efficiency of the extracts. Another explanation for the results is the solubility of the particles dissolved in the solution [4]. These results are consistent with those reported in previous research, i.e., that the (%) EE was increased with increased total tannin recovery at varying concentration ratios of wall materials [28]. Recent evidence suggests that the combination of maltodextrin and gum arabic produced better encapsulation of phenolic and antioxidant capacity, with higher EE values for use in feed additives [28,46,47]. Additionally, numerous studies have reported that the formulation of microencapsulation of bioactive compounds using different encapsulant materials, including whey protein, sodium caseinate, lecithin, rapeseed oil, pectin, and cholesterol, via spray-drying technique was the most effective method for delivery into fermented milk products [48,49]. 

It was difficult to compare the outcomes owing to the novelty of using *Cannabis* species and *Mytragyna speiosa* leaves as the core materials in encapsulation with the spray-drying technique. However, Kurek et al. [2] reported that spray-dried microcapsules prepared from hempseed oil by employing a combination of maltodextrin with hempseed/leaf extract resulted in lowest EE, with 20% oil content (37.12%), whereas the highest EE resulted from the combination of rice protein and maltodextrin, with 10% oil content (79.37%). Additionally, in a study on the use of pea protein and pectin for spray-dried encapsulation of polyunsaturated fatty-acid-rich oil, Aberkane et al. [50] concluded that pea protein was not a suitable wall material to increase encapsulation efficiency and minimize lipid oxidation.

### 3.3. Microstructure and Surface Morphology of Dried Extracts and Chitosan Microcapsules

The aim of this phase of the study was to investigate the microstructure, size, shape, and surface morphology of extracts from the leaves of the three investigated plants after microwave extraction and comparison of the changes in the extract characteristics through encapsulation using spray-drying technique under FE-SEM, as shown in Figure 2a–f. FE-SEM micrographs of dried extracts and unencapsulated extracts obtained from the three plants revealed that their particles had irregular shapes, both circular and square. The surface morphology was smooth, shining, or glossy, with grooves and multiple impregnations. The diameter of the circular particles ranged from 0.70 to 1.98 μm, whereas that of the square particles ranged from 22 to 65 μm (Figure 2a–c). A number of studies have reported that untreated hemp hurd biomass (*Cannabis sativa*) appears rough under SEM, with sharp edges and grooves; however, pretreatments lead to significant and numerous morphological alterations, indicating significant damage following pretreatment [51]. Other authors reported that extracted wet pectin had wrinkles on the surface and a smooth, compact nanostructure compared to extracted pectin dried in an oven, with mound-shaped pellets formed on the smooth pectin surface, indicating that drying resulted in some damage and swelling effects on the pectin structure [52].

As shown in Figure 2d–f the chitosan microcapsules of bioactive compound particles were formulated by chitosan-Tween 80 (Polysorbate 80) with characteristics of ionic interaction between a positively charged amino group of chitosan and a negatively charged Tween 80 as a surfactant. The surface morphologies of chitosan microparticles were completely spherical in shape, with smooth and/or rough surfaces interspersed with porous surrounding particle spheres. The microencapsulated *C. sativa* presented with numerous particles ranging in size from 0.72 to 9.49 μm in diameter, which is a particle size similar to that of microencapsulated *C. indica*, mostly in the range of 1.45 to 9.69 μm, whereas the majority of microencapsulated *M. speiosa* particles had a diameter of 1.52 to 11.0 μm. Moreover, the outer membrane of these microencapsulated particles was particularly interesting, with clear modifications evidencing adherence of the wall materials to the bioactive compound particles in the form of several layers of coating (Figure 2d–f). Researchers have mostly reported that the encapsulation of bioactive compounds result in a uniform, spherical matrix with flakes when formulated with chitosan as wall material using the spray-drying technique [2,3,53].

The results of this study also indicate the safety of microencapsulated particle formation techniques without organic solvent additives. Therefore, the proposed method is safe for animal health and can be used as a supplement for feed additive technology.

## 4. Conclusions

The bioactive compounds of the leaves of *M. speiosa, C. indica*, and *C. sativa* obtained in the present study using microwave extraction could be effectively retained by microencapsulation with chitosan by spray-drying technique. A high concentration of bioactive components containment, especially TPC, TFC, and antioxidant capacity, was achieved in these microcapsules with high-quality bioactive compounds.

## Figures and Tables

**Figure 1 antioxidants-11-02103-f001:**
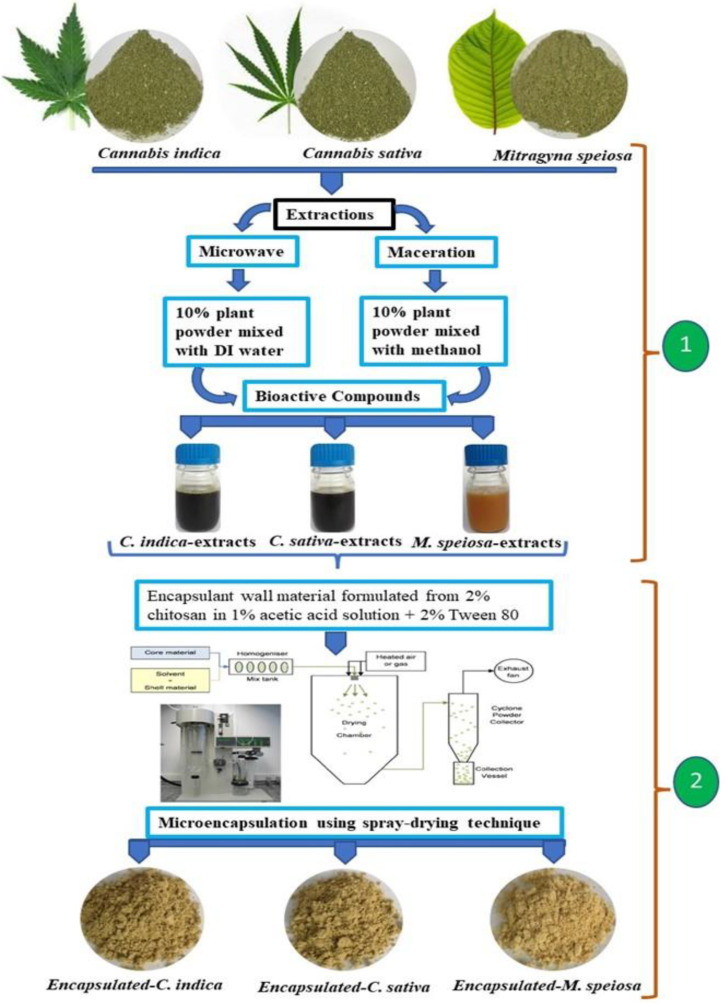
Procedure for bioactive compound extraction (1) and the microencapsulation formulation processing steps (2).

**Figure 2 antioxidants-11-02103-f002:**
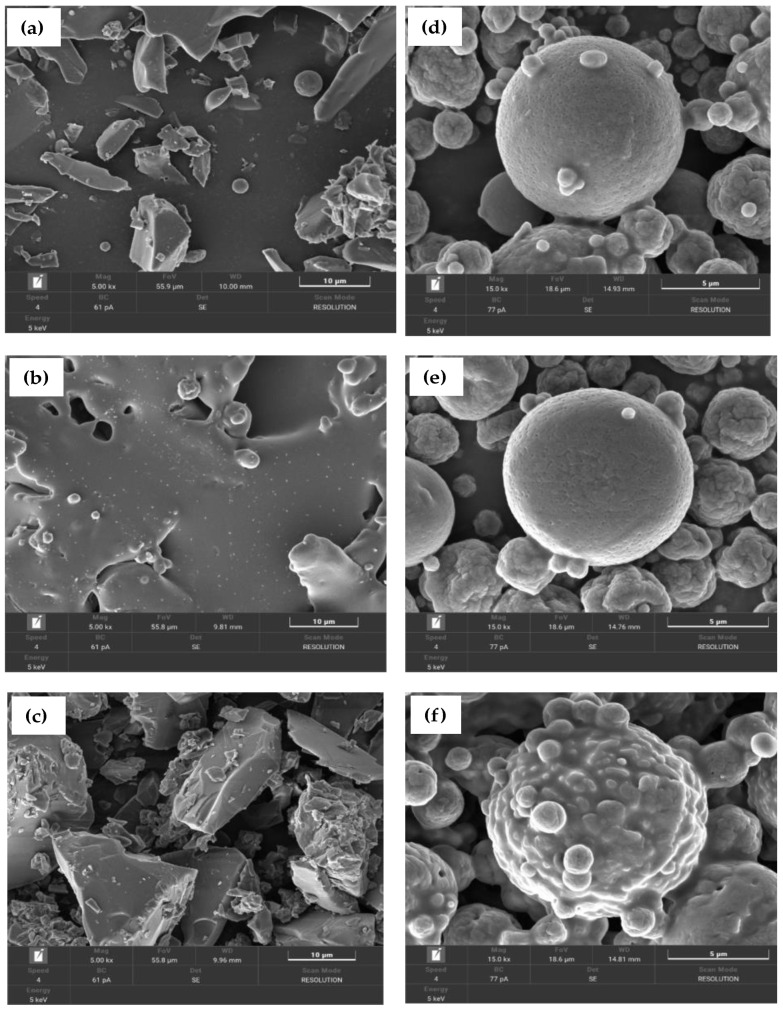
FE-SEM images of microencapsulated bioactive compound nanoparticles (components of 2% chitosan combined with 2% Tween 80; (**a**,**d**): unencapsulated and encapsulated *C. sativa*, (**b**,**e**): unencapsulated and encapsulated *C. indica*; (**c**,**f**): unencapsulated and encapsulated *M. speiosa*.

**Table 1 antioxidants-11-02103-t001:** The bioactive and antioxidative values of three plant leaves obtained via different extraction methods.

Extraction Method	Bioactive Compounds		Antioxidant Capacity		
	TPC (mg GAE/g DM)	TFC (mg QUE/g DM)	DPPH Inhibition (%)	ABTS Inhibition (%)	FRAP Capacity (mg TROE/g DM)
**Microwave extraction**					
*C. sativa*	171.7 ± 0.4 ^c^	66.2 ± 4.7 ^c^	35.0 ± 6.4 ^b^	43.1 ± 2.3 ^b^	14.0 ± 0.7 ^c^
*C. indica*	218.9 ± 0.9 ^b^	88.6 ± 3.4 ^b^	39.3 ± 6.1 ^b^	94.6 ± 1.0 ^a^	23.7 ± 0.5 ^b^
*M. speiosa*	306.9 ± 0.3 ^a^	119.2 ± 5.2 ^a^	91.4 ± 0.5 ^a^	95.3 ± 0.4 ^a^	39.0 ± 0.1 ^a^
**Maceration extraction**					
*C. sativa*	11.3 ± 0.6 ^c^	7.1 ± 2.4 ^b^	24.7 ± 0.3 ^c^	3.7 ± 0.4 ^c^	7.7 ± 1.7 ^c^
*C. indica*	17.7 ± 1.1 ^b^	10.5 ± 1.9 ^a, b^	38.0 ± 0.5 ^b^	20.6 ± 0.6 ^b^	14.4 ± 0.3 ^b^
*M. speiosa*	21.7 ± 0.6 ^a^	13.7 ± 2.7 ^a^	91.3 ± 0.0 ^a^	88.9 ± 0.1 ^a^	39.4 ± 0.1 ^a^

Values are expressed as the mean ± SD (*n* = 3). For each column, the letters ^a^, ^b^, and ^c^ indicate significant differences (*p* < 0.05). TPC, total polyphenolic content; TFC, total flavonoid content; DPPH inhibition, DPPH radical-scavenging activity; ABTS inhibition, ABTS radical-scavenging activity; FRAP capacity, ferrous ion reducing power; GAE, gallic acid equivalent; QUE, quercetin equivalent; TROE, Trolox equivalent.

**Table 2 antioxidants-11-02103-t002:** Bioactive and antioxidative values, as well as the encapsulation efficiency of chitosan-encapsulated compounds obtained from three plant leaves using the spray-drying technique.

Formulations	Bioactive Compound		Antioxidant Capacity			Encapsulation Efficiency (%)
	TPC (mg GAE/g DM)	TFC (mg QUE/g DM)	DPPH Inhibition (%)	ABTS Inhibition (%)	FRAP Capacity (mg TROE/g DM)	
Encapsulated-*C. sativa*	240.0 ± 6.6 ^c^	22.6 ± 2.7 ^c^	69.9 ± 0.7 ^c^	53.0 ± 0.4 ^a^	17.2 ± 1.1 ^b^	54.6 ± 10.7 ^c^
Encapsulated-*C. indica*	266.1 ± 6.8 ^b^	69.8 ± 3.0 ^b^	83.5 ± 1.0 ^b^	84.3 ± 0.9 ^b^	30.9 ± 0.7 ^a, b^	82.3 ± 7.6 ^b^
Encapsulated-*M. speiosa*	307.8 ± 6.4 ^a^	105.3 ± 3.6 ^a^	94.8 ± 0.4 ^a^	90.3 ± 1.2 ^a^	34.4 ± 0.7 ^a^	99.7 ± 4.1 ^a^

Values are expressed as the mean ± SD (*n* = 3). For each column, the letters ^a^, ^b^, and ^c^ indicate significant differences (*p* < 0.05). TPC, total polyphenolic content; TFC, total flavonoid content; DPPH inhibition, DPPH radical-scavenging activity; ABTS inhibition, ABTS radical-scavenging activity; FRAP capacity, ferrous ion reducing power; GAE, gallic acid equivalent; QUE, quercetin equivalent; TROE, Trolox equivalent.

## Data Availability

The data is contained within the article.

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
