# Peer review of "Extraction, Characterization, and Chitosan Microencapsulation of Bioactive Compounds from Cannabis sativa L., Cannabis indica L., and Mitragyna speiosa K."

_antioxidants, 2022, doi:10.3390/antiox11112103_

Round 1
Reviewer 1 Report
The Authors report on a chitosan-microencapsulation technology to encapsulate plant extracts in order to improve efficiency. As expected, higher polyphenolic content and antioxidant capacity were detected in the microencapsulated particles.
My concerns are below:
- The AA report an array of methods for total polyphenolic, total flavonoid contents, and antioxidant capacity. I would recommend a qualitative/quantitative identification of the polyphenols present in the extracts with and without the encapsulation process. This will provide insights on the distribution of the phytochemicals in the extracts.
- Table 1 reports the bioactive compound components of three plant leaves obtained from different extraction methods, specifically microwave extraction and maceration extraction. Microwave extraction provided higher yields. Could the AA identify the bioactive affected by microwave extraction?
- The AA report significant differences in (%) EE based on the amount of TPC contents after being encapsulated (Table 2). Could they speculate on the reasons behind those observed differences amongst the three extracts? Is it to do with the phytochemicals structure and composition?
Author Response
Dear Editor-in-Chief and Reviewers,
We are very delightful to receive all the comments and suggestions made by the Section Editor and reviewers. Above all, the authors felt that all points made were very useful and we have incorporated most of the corrections, as suggested in order to make the manuscript ready for possible publication in Antioxidants. All those corrected and modified appeared in yellow colored highlight.
We declare that all changes of authors list are well accepted and all co-authors are aware of this.
With the above information we would like to resubmit our paper for your kind considerations for a possible publication in Antioxidants.
We again wish to sincerely thank you very much for your kind attention and support.
Sincerely Yours,
Prof. Dr. Metha Wanapat
On behalf of all authors

Reviewer 2 Report
The manuscript covers the effective method of antioxidant substances obtaining from plant material and preparing them for convenient use as a feed additive. There are some issues to be clarified. Please find the following remarks helping to improve the manuscript.
1. The title of the manuscript should be revised. The expression “for Effective Future Use of Animal Feed Additives” should be removed. Moreover, the plants should be mentioned in the title to be more informative.
2. Abstract: The aim of the study should be precisely stated. I also suggest presenting the numerical data illustrating the results of the study. What was the antioxidant value of the encapsulated plant derivatives?
3. Introduction: What was the reason for choosing these specific plant materials? What is the nutritional value of these plant derivatives? Please introduce the data concerning the plant material and justify the use of Cannabis sativa L., Cannabis indica L., and Mitragyna speiosa K. as animal fed additives. Please rephrase “the morphological characteristics under the scanning electron microscope (SEM)” (lines 79-80), specifying the type of analysis and the parameters of chitosan-microcapsules of bioactive compounds.
4. Lines 115-129: Please add the type and brand of spectrophotometers used in the study.
5. Lines 145-146: Figure 1 caption should be rephrased e.g. Procedure of the ….
6. Lines 147-152: Please add the type and brand of spectrophotometers used in the study. Please provide a detailed procedure. The material “before” the encapsulation serves the only plant material and “after” the encapsulation the mass is fortified with the chitosan. How can you value the efficiency? In my opinion we should consider the amount of the extracts per the mass unit of the microcapsules. The sentence “All the experiments were carried out in triplicates and expressed by the following equation” should be also rephrased.
7. Lines 153-156: The title “2.7. Morphological characterization” should be precise e.g. 2.7. Morphological characterization of microcapsules. The parameters of microcapsules scrutinised should be listed in methodology.
8. Line 159: Please remove “using the excel program”.
9. Please rephrase “TPC content” and “TFC content” for TPC (TPC = Total Phenolic Content) and TFC (TFC = Total Flavonoid Content) within the text.
10. Please note that “TPC extraction” expression cannot be used (e.g. line 187) as TPC is the level of total phenolic compounds.
11. Lines 174-177: Something is missing in the sentence “Thus, the extracts containing the phytochemical contents of TPC, TFC, and antioxidative activities such as DPPH radical scavenging inhibition, ABTS radical scavenging inhibition, and FRAP reducing power capacity, respectively.” Please rephrase.
12. Lines 184-186: Something is missing in the sentence “According to Drini´c et al. [13], who have reported the optimization of MAE operating conditions during polyphenolic contents and antioxidant capacity extraction from hemp with ethanol to water at different ratios.” Please rephrase.
13. Lines 232-233: Table 1 caption is improper. The values presented in Table 1 are the not the “The bioactive compound components”. There are the measures of the antioxidant value of the extract but not compounds.
14. The discussion of the results lacks the comparison of antioxidant activity of extracts Cannabis sativa L., Cannabis indica L., and Mitragyna speiosa K. with literature data. There is no a scientific background of the antioxidant activity of the plant extracts under research.
15. Lines 239-265: “3.2. Encapsulation efficiency of microencapsulated-bioactive compounds” paragraph is not clear for me. I have doubts about the designation of encapsulation efficiency (see methodological remarks). Moreover, the results discussion does not focus on the research and experiment design. The results should be recalculated and the section totally rearranged.
16. Table 2 caption is missing.
17. Lines 266-286: Please show the aim of the visualisation of the microcapsules. This part of the manuscript should be combined with the previous paragraph and placed under a different title. How the Authors explain the big differences in the sizes of extract particles and microcapsules particles?
18. Lines 275-276: Please revise the sentence ” The diameter of the circular particles ranged from 0.70 to 1.98 μm, whereas the diameters of the square particles varied between 22 and 65 μm”, especially in terms of the square particles.
19. Lines 280-282: “Moreover, the outer membrane of these microencapsulated particles was particularly interesting of clear modifications and evidence of adherence of the wall materials to the bioactive compound particles in the form of several layers of coating.” What is the meaning of “outer membrane” and “wall materials” here? Please specify “several layers of coating” in Fig. 2.
20. Lines 291-295: The interpretation of the results goes too far and is not based on the results of the research.
21. The references formatting does not meet the Journal requirements.
Author Response

(The authors gave the same response as above.)

Round 2
Reviewer 1 Report
The Authors have addressed the comments appropriately.
Author Response
Dear Editor-in-Chief and Reviewers,
We are very delightful to receive all the comments and suggestions made by the Section Editor and reviewers. Above all, the authors felt that all points made were very useful and we have incorporated most of the corrections, as suggested in order to make the manuscript ready for possible publication in Antioxidants. All those corrected and modified appeared in yellow colored highlight.
We declare that all changes of authors list are well accepted and all co-authors are aware of this.
With the above information we would like to resubmit our paper for your kind considerations for a possible publication in Antioxidants.
We again wish to sincerely thank you very much for your kind attention and support.
Reviewer 2 Report
However, the Authors considered majority of my remarks, but I still think some of the issues should be improved. Please find the following remarks I hope helping the mastering the manuscript:
1. Abstract: The aim of the study was specified but I do not think “to develop the extraction method to enhance bioactive compounds and retain their compounds using the microencapsulation technique for use as rumen enhancers” is proper. The extraction methods are rather not innovative and well known, so they cannot be stated as being developed. How were the enhancement of bioactive compounds realised? Also placing “for use as rumen enhancers” needs to be verified in the manuscript if the Authors state it in the aim of the study. Under presenting the numerical data illustrating the results of the study I meant the antioxidant value of the encapsulated plant derivatives, so please supply the relevant data.
2. Introduction: Please remove “sprayed with gold for measurements “specifying the type of analysis and the parameters of chitosan-microcapsules of bioactive compounds.
3. Lines 167-177: I fully accept the procedure of TPC estimation but in the “Encapsulation efficiency” procedure the chitosan counts in the mass of the measured material. Has chitosan been included in the mass of the determined sample? Pease note again: the material “before” the encapsulation serves the only plant material and “after” the encapsulation the mass is fortified with the chitosan. How can you value the efficiency? The text added is not consistent and must be rephrased. Please remove repetition of the Folin-Ciocateau method expression, which was specified in TPC assessment methodology.
4. Lines 181: What do you mean under “outer membrane”? Was it the surface of the microcapsule? Please correct.
5. I was asking for rephrasing “TPC content” and “TFC content” for TPC (TPC = Total Phenolic Content) and TFC (TFC = Total Flavonoid Content) within the text. Both abbreviations contain the word “content”, so “TPC content and TFC content” should be replaced by e.g., “TPC” or “TFC”. Please correct all these expressions within the text of the manuscript.
6. Lines 201-203: The sentence “Thus, the extracts containing the phytochemical contents of TPC, TFC, and antioxidative activities such as DPPH radical scavenging inhibition, ABTS radical scavenging inhibition, and FRAP reducing power capacity were carried out, respectively.” Still needs to be rephrased.
7. Lines 214-216: Something is still missing in the sentence “According to Drinić et al. [137], who have reported the optimization of MAE operating conditions during polyphenolic contents and antioxidant capacity extraction from hemp by using different ratios of ethanol to water.” Please rephrase. Maybe it should be combined with the next sentence?
8. The discussion of the results lacks the comparison of antioxidant activity of extracts Cannabis sativa L., Cannabis indica L., and Mitragyna speiosa K. with literature data. There is not a scientific background of the antioxidant activity of the plant extracts under research.
9. Lines 269-303: I still have doubts about the designation of encapsulation efficiency (see methodological remarks). The results should be recalculated.
10. Table 2 caption should be thoroughly corrected.
11. Lines 315-325: “3.3. Microstructure and surface morphology” title is not complete, please correct. Please combine 3.3.1 and 3.3.2 paragraphs and discuss your finding with the literature.
12. Lines 359-367: The conclusions were not corrected. The interpretation of the results goes too far and is not based on the results of the research.
Author Response
Dear Editor-in-Chief and Reviewers,
We are very delightful to receive all the comments and suggestions made by the Section Editor and Reviewers. Above all, the authors felt that all points made were very useful and we have incorporated most of the corrections, as suggested in order to make the manuscript ready for possible publication in Antioxidants. All those corrected and modified appeared in yellow colored highlight.
We declare that all changes of authors list are well accepted and all co-authors are aware of this.
With the above information we would like to resubmit our paper for your kind considerations for a possible publication in Antioxidants.
We again wish to sincerely thank you very much for your kind attention and support.
